# Uncertainty Analysis of Business Interruption Losses in the Philippines Due to the COVID-19 Pandemic

**Joost R. Santos** [1,*], **John Frederick D. Tapia** [2], **Albert Lamberte** [3], **Christine Alyssa Solis** [3], **Raymond R. Tan** [2], **Kathleen B. Aviso** [2] **and Krista Danielle S. Yu** [3]

1   Department of Engineering Management and Systems Engineering, George Washington University, Washington, DC 20052, USA
2   Department of Chemical Engineering, De La Salle University, Manila 0922, Philippines
3   School of Economics, De La Salle University, Manila 0922, Philippines
\*   Correspondence: joost@gwu.edu

**Abstract:** In this study, we utilize an input–output (I–O) model to perform an ex-post analysis of the COVID-19 pandemic workforce disruptions in the Philippines. Unlike most disasters that debilitate physical infrastructure systems, the impact of disease pandemics like COVID-19 is mostly concentrated on the workforce. Workforce availability was adversely affected by lockdowns as well as by actual illness. The approach in this paper is to use Philippine I–O data for multiple years and generate Dirichlet probability distributions for the Leontief requirements matrix (i.e., the normalized sectoral transactions matrix) to address uncertainties in the parameters. Then, we estimated the workforce dependency ratio based on a literature survey and then computed the resilience index in each economic sector. For example, sectors that depend heavily on the physical presence of their workforce (e.g., construction, agriculture, manufacturing) incur more opportunity losses compared to sectors where workforce can telework (e.g., online retail, education, business process outsourcing). Our study estimated the 50th percentile economic losses in the range of PhP 3.3 trillion (with telework) to PhP 4.8 trillion (without telework), which is consistent with independently published reports. The study provides insights into the direct and indirect economic impacts of workforce disruptions in emerging economies and will contribute to the general domain of disaster risk management.

**Keywords:** Leontief model; Dirichlet distribution; uncertainty analysis; workforce; disaster; pandemic

## 1. Introduction

Outbreaks of endemic diseases occur seasonally. Nonetheless, albeit relatively rare, severe epidemics that encompass large geographic areas and with high transmission rates—known as pandemics—have been well documented in the past. Early in the 20th century, the 1918 H1N1 pandemic nicknamed as the "Spanish Flu" became one of the deadliest disasters in modern times, killing more than 50 million people worldwide (Taubenberger and Morens 2006). Furthermore, mild to moderate pandemics occurred thereafter such as the 1952 H2N2, 1968 H3N2, and the more recent 2009 H1N1pdm2009 (Centers for Disease Control and Prevention 2018).

In 2007, the World Health Organization (WHO) issued a statement that emphasized the urgency of a coordinated effort amongst the global scientific community to prepare for and combat the risks associated with new infections (WHO 2007). Prior to the COVID-19 pandemic[1], the current generation had not experienced a disaster of similar magnitude as the 1918 Spanish Flu. Indeed, we became firsthand witnesses to the complex consequences of COVID-19 such as mortalities, exceedance of healthcare capacity, and economic collapse, which punctuated the clear and present threat of pandemics. This has resulted in port closures and lockdowns, which have caused major supply chain disruptions whose impacts cross geographical boundaries (Yu and Aviso 2020). Furthermore, the transmission rate

and severity of viral outbreaks are fraught with uncertainties. Increased burden on the healthcare sector arises as frontliners have heightened exposure that can worsen the sector's inoperability (El Haimar and Santos 2015). Coupled with their ability to mutate and evade vaccination efforts, the recovery from a pandemic can be a complex and daunting effort (Orsi and Santos 2010).

For novel diseases, non-pharmaceutical intervention (NPI) measures are implemented in the absence of vaccines. The three major categories of NPI measures are containment, suppression, and mitigation. Containment is the ability to detect the infected individuals and separate them from the general population. Suppression, on the other hand, refers to measures that can reduce the reproduction number[2] to a value lower than 1, which may be achieved via business and school closures, travel restrictions, and lockdowns, among others. Finally, mitigation measures are the ones directly associated with the "flatten the curve" concept (Centers for Disease Control and Prevention 2007, 2017), which are also referred to as the 3Ws of mitigation—wash your hand, watch your distance, wear your personal protective equipment (PPE). Many studies have assessed both the efficacy of various NPI measures in reducing and delaying the impact of infections, as well as the unwanted adverse effects associated with mental health and infringement of personal liberties (Ferguson et al. 2020; Huzar 2020; James et al. 2020; Pueyo 2020).

Unlike most disasters where the direct damage is on physical infrastructure and the natural environment, the impact of pandemics is concentrated on people. The workforce is the lifeblood of any society, and massive disruptions to the workforce will immediately translate to the inoperability of critical infrastructure systems (Santos et al. 2014a). Hence, although pandemics do not render direct damage to physical systems, the consequences can be as debilitating as manmade disasters (Santos 2020a). In this paper, the focus of the analysis is on workforce disruptions attributable to NPI measures like lockdowns, business closures, and travel bans. Indeed, disasters like pandemics can lead to "forced" absenteeism because physical access to places of work is curtailed, or worse, completely restricted. In addition, workforce absenteeism can also be caused by workers getting sick themselves or providing care to sick family members, thus, requiring them to quarantine. Because of the significant impact triggered by workforce disruptions, government agencies such as the US Department of Labor (2020) have proposed policy recommendations to help reduce the subsequent economic losses. For example, teleworking (or "working from home") has provided an opportunity for sectors to continue their operations amid the pandemic. The availability of infrastructure and the nature of jobs allows for teleworking. Furthermore, some sectors have creatively leveraged information technology to maintain their capability to deliver goods and services virtually to lessen the impact of workforce disruptions. The pandemic has fast-tracked the adoption of teleworking across different jobs and sectors. Prior to the pandemic, the percentage of employees who were able to telework ranged from only 2 to 20 percent varying across countries, occupations, and sectors (Messenger 2019). However, the onset of the pandemic has led to a surge in teleworking. In the EU, 39.6% of paid work was conducted from the employees' homes and 46% had never teleworked prior to the pandemic (Eurofound 2020).

The primary contribution of this article is to model and evaluate the positive impact of teleworking and their associated economic benefits. In particular, the focus of this paper is to evaluate the benefits of teleworking on the continuity of operations across various sectors. The subsequent sections of this article are organized as follows. In Section 2, the methodology for modeling of workforce disruptions using input–output (I–O) modeling is discussed, as well as the use of Dirichlet distributions to model the uncertainty of the I–O parameters. The relevant data sources are also discussed in Section 2. In Section 3, we perform simulations of various telework scenarios and evaluate the resulting distribution of economic losses. The results are further discussed in Section 4. Finally, the conclusions of the paper and other reflections for future consideration are presented in Section 5.

## 2. Materials and Methods

### 2.1. Overview of Pandemic Impact Modeling

Pandemic and disaster risk management encompass a wide range of spatial and temporal dimensions (Okuyama and Santos 2014). However, identifying the optimal model for disaster risk analysis tends to be constrained by the availability of quality data, limitations of quantitative techniques, and interpretation of the results (Albala-Bertrand 1993). Santos et al. (2014a) introduced the WEIGHT framework (Workforce, Economy, Infrastructure, Geography, Hierarchy, and Time) to examine critical factors affecting disaster risk assessment and management. As the nature of international trade entails an interconnected economy, countries have become more vulnerable to foreign shocks.

COVID-19 has brought about global economic contraction due to a decline in consumption, disrupted global supply chain and production, business closures, and loss of work, causing an increase in the number of people living in extreme poverty with 3.3 billion workers worldwide at risk of losing their jobs (WHO 2020). Haleem et al. (2020) categorized the impact of the pandemic into three areas: healthcare, economic, and social. Other than the direct effect of the morbidity and mortality of the disease, the medical system is overstretched and highly burdened due to the overwhelming number of cases within a short period of time. Non-COVID patients have become neglected as healthcare professionals become understaffed due to exposure or quarantine measures. The medical supply chain was also disrupted due to stringent lockdown measures implemented both domestically and globally. The impact on the economy manifests in the reduced consumption of goods and disrupted global supply chain network. Lastly, the disruption to the social aspect includes the imposed social distancing measures, leading to the closure of social spaces such as restaurants, malls, entertainment areas, travel restrictions, and stalled service sector.

Previous studies have examined the impact of pandemics, and provided optimal intervention strategies that minimize economic losses, yet save thousands of lives. Ginsberg et al. (2009) emphasized the need for early detection and response to mitigate the impact of both season and pandemic influenza. Eichenbaum et al. (2021) identified a trade-off between negative short-run economic outlook and the health of the population, where ending containment early (before 44 weeks) can create a surge in infection rates while starting containment too late (after 33 weeks) can cause the death of up to 0.4% of the initial population. Ferguson et al. (2006) determined that intervention measures such as rapid case isolation can reduce cumulative attack rates by 7% (if 90% of cases are isolated), while the implementation of border controls can only delay the spread by three to six weeks if imported infections were reduced by over 99%. Lastly, a combination of personal behavioral change and government intervention would be the optimal strategy to mitigate the spread of the pandemic, emphasizing the role of the individual to conduct self-isolation, social distancing, and to seek remote medical assistance (Anderson et al. 2020). Governments may also impose travel restrictions and conduct rigorous contact tracing to ensure spread containment.

Dingel and Neiman (2020) argued that low-income economies tend to have fewer jobs that can be conducted remotely or at home. Governments implemented NPI measures to mitigate the spread of infection before the use of vaccines became widespread. NPI measures are done through the combination of containment, suppression, and mitigation measures (Santos 2020a). However, such measures may result in forced absenteeism because of mobility restrictions such as lockdowns or lack of transportation (Santos 2020b). Sectors reliant on migrant workers were hit particularly hard (Foong et al. 2022).

Several studies on post-disaster workforce unavailability exposed the adverse economic impact of disruptions. Orsi and Santos (2010) translated the effect of workforce unavailability as a measure of sector productivity disruption in the state of Virginia. Using three potential attack rates (15%, 25%, and 35%), the expected value of economic losses would be USD 4.6 billion, USD 7.7 billion, and USD 10.8 billion, respectively. Chen et al. (2021) examined the trade-off between economic losses due to inoperability, lockdown duration, and averted infections and deaths. In an unmitigated scenario (no lockdown or

NPI measures), economic losses were highly reduced but resulted in 100 million infections and a hundred thousand mortalities. On the other hand, a lockdown scenario of 45 days with 90% compliance rate will result in an economic loss of about USD 3.4 trillion but will save over 110,000 lives and mitigate the infection of 115 million individuals.

The unpredictable nature of the pandemic has led to governments implementing varying levels of mobility restrictions over an extended period of time that resulted in persistent inoperability in the economy (Yu et al. 2020b). The COVID-19 pandemic has forced an increased adoption of alternative working arrangements globally to reduce the spread of the virus. In the United States, 35.2% of the workforce were working from home by May 2020 in contrast to 8.2% in February (Bick et al. 2020). There was also an increase in work from home arrangements in the EU. Eurofound (2020) launched a survey and found that more than 48% of its respondents who were categorized as employees were now partially working from home, while 34% worked exclusively from home. In the Philippines, work-from-home (WFH), or telework arrangements were adapted by businesses if their line of work allows their employees to do so. Gaduena et al. (2022) estimated that only 25.7% of occupations in the Philippines are teleworkable. Although there was a swift adoption of such measure to those that have the capability to do so, some jobs cannot be performed remotely (Gaduena et al. 2022). Further, Gaduena et al. (2022) identified that most of the occupations that can support teleworkable jobs are typically found in sectors that have a low share of the country's total employment (i.e., sectors that require at least undergraduate degrees). The Organization for Economic Co-operation and Development (OECD) (OECD 2020) also states that most teleworkable jobs require highly skilled workers. Generalao (2021) also made a similar argument, such that industries that employ more educated individuals can experience the benefits of technological advances. On the other hand, those occupations that require lower levels of education, such as low- and medium-skilled laborers, are considered non-teleworkable (Gaduena et al. 2022; OECD 2020). Telework arrangements brought about an unprecedented demand surge for electronics. A prolonged shortage in essential technology metals will affect production of equipment necessary for telework (Yu et al. 2020a). Thus, persistent supply chain disruptions can affect teleworkability in the long run.

### 2.2. Economic Input–Output Model

The input–output (I–O) model provides a mathematical framework to model the interdependencies among economic sectors that comprise an economy (Leontief 1936). It has been used for numerous applications involving analysis and forecasting and is considered as a standard economic tool in most countries (Miller and Blair 2009). I–O models use systems of linear equations to capture the network-like structure of interlinked economic structures. This feature makes this approach particularly useful for studying the changes in production and consumption of interdependent economic sectors as a result of network ripple effects. The basic I-O model can also serve as the basis for more sophisticated models. Examples include mathematical programming extensions as well as computable general equilibrium (CGE) models that relax the strict assumption of the Leontief production function (Miller and Blair 2009).

The basic I–O model may be written as follows:

$$x = Ax + y \tag{1}$$

where $A$ is the matrix of technical coefficients, $x$ is the vector of total sector outputs, and $y$ is the vector of sector final demands. Equation (1) shows that the total output must meet the combined intermediate and final demands. In conventional I–O applications, these flows are given in terms of monetary value. The coefficients of $A$ reflect the average state of production technology used in the system's sectors. Their proportions are assumed to be fixed to reflect a Leontief production function that does not allow for substitution among inputs. This restrictive assumption is generally considered as acceptable for typical

applications of I–O models, which are understood to provide snapshots of the state of the economies that they represent. Rearranging Equation (1) gives:

$$x = (I - A)^{-1}y \tag{2}$$

where $(I - A)^{-1}$ is the Leontief inverse. In this form, the I–O model can be readily used to compute production levels associated with any given final demand scenario.

The basic I–O model has also been extended to account for the flow of inoperability through economic or infrastructure networks (Haimes and Jiang 2001). Inoperability is a dimensionless metric of the degree of system failure, with a value ranging from 0 for a system at normal state to a value of 1 for a system in a state of total failure. The original physical definition of inoperability was modified by Santos and Haimes (2004) to allow inoperability I–O models (IIM) to be calibrated using published economic data. In such applications, inoperability quantifies the relative or fractional drop in economic activity (Santos 2006). The IIM formulation based on demand disruption is shown in Equation (3):

$$q = A^*q + c^* \tag{3}$$

where $A^*$ is the interdependency matrix, $c^*$ is the sector demand disruption vector, and $q$ is the sector inoperability vector. Matrix $A^*$ can be determined from the baseline state of the economy being analyzed (Santos and Haimes 2004). Note that this model is structurally analogous to the basic I–O model, and can also be rearranged as follows:

$$q = (I - A^*)^{-1}c^* \tag{4}$$

where $(I - A^*)^{-1}$ is analogous to the Leontief inverse, and acts as a multiplier that quantifies the amplification effect of an economic system on an initial input disruption ($c^*$) to yield the sectoral interoperability ($q$). This demand-reduction form of the IIM model was first used to analyze the economic ripple effects of the September 11 terrorist attacks in the United States (Santos and Haimes 2004). A two-part review on IIM was subsequently published, focusing on fundamentals (Haimes et al. 2005a) and applications (Haimes et al. 2005b), respectively. In addition to these early uses of IIM, more recent examples of disruptive events modelled with IIM in the previous decade include earthquakes and tsunamis (MacKenzie et al. 2012), cyberterrorist attacks (Ali and Santos 2015), droughts (Santos et al. 2014b), crop failure due to infestation (Aviso et al. 2015), and influenza epidemics (Santos et al. 2013).

The basic I–O model and IIM can be used in parallel to analyze the effects of disasters and other disruptive events, since they reflect different aspects of the system. These two metrics thus generally lead to different priority rankings of economic sectors (Santos et al. 2013). For example, large economic sectors can experience high levels of economic loss even at relatively low inoperability. Conversely, small sectors may experience seemingly small economic losses even at high levels of inoperability. This distinction has important uses for supporting policy decisions. Economic loss estimates from the basic I–O model can be interpreted directly in terms of social welfare implications. On the other hand, inoperability reflects the actual extent of loss suffered by a given sector relative to its normal state. Since economic sectors are the aggregation of multiple private businesses, inoperability can more accurately reflect the collective damage done to these entities (e.g., a wave of bankruptcies in a given sector may hamper its ability to recover from a transient crisis). The inoperability metric can also be linked to shifts in economic structure leading to improved resilience (Okuyama and Yu 2019).

### 2.3. Uncertainty Analysis

In the field of probabilistic risk analysis (PRA), it is commonplace to estimate a consequence metric from a disaster as a range or as a point estimate with an associated confidence interval. Indeed, Kaplan and Garrick (1981) formulated the triplet of questions

in PRA that explicitly contains the element of uncertainty. These questions are: (i) "What can go wrong?", (ii) "What is the likelihood?", and (iii) "What are the consequences?".

Uncertainty analysis is a broad subject area, encompassing the two general categories of uncertainty—aleatory (statistical variability or randomness) and epistemic (lack of knowledge). In this paper, the emphasis is on the uncertainty of the model parameters of the I–O model, which was discussed in the previous section. In particular, we will model the effects of the I–O technical coefficient matrix (denoted by $A$ in Equation (1)).

In the domain of I–O modeling and analysis, several papers have emphasized the importance of conducting uncertainty analysis. Rose (2004) argues that the use of a deterministic point estimate to discuss the result of an analysis reflects the exaggeration of certainty and will most likely be erroneous. Furthermore, quoting from Gerking (1976): "It is commonplace in I–O analysis to interpret [the static I–O equation] as a deterministic or exact forecast. However, to contend that this forecast will come true with certainty is optimism to a fault". As a corollary to such arguments, deterministic modeling estimates may lead to faulty policies due to the illusion of certainty that they create.

This paper proposes the use of Dirichlet distribution to model the uncertainties associated with the elements of the I–O matrix. The Dirichlet distribution can be thought of as an extension of the Beta distribution. When modeling a set of variable proportions that add up to 1, the appropriate distribution to use is the Dirichlet distribution. As a matter of fact, several papers have explicitly used the Beta and Dirichlet distributions to model uncertainty of the I–O coefficients (see, for examples, Jansen 1994; Ten Raa and Rueda-Cantuche 2007; Dietzenbacher 2006).

The probability density function of the Beta distribution is as follows, where $x$ is the random variable, and the Greek letters are the standard notation for the Beta distribution parameters:

$$f(x; \alpha, \beta) = \frac{\Gamma(\alpha+\beta)}{\Gamma(\alpha)\Gamma(\beta)} x^{\alpha-1}(1-x)^{\beta-1}$$
$$0 \leq x \leq 1 \text{ and } \alpha, \beta > 0 \tag{5}$$

The Beta distribution is commonly used to model a proportion. For example, if we look at the proportion of the manufacturing sector with respect to the total gross domestic product (GDP), such proportion can change from year to year and hence can be modeled as a Beta distribution. The Beta distribution can only handle one proportion at a time as a random variable. There are cases where it is desirable to model multiple proportions that add up to one. For example, consider the Philippine I–O table where each column comprises of 16 sectors + value added (see Supplementary Materials). Each column can be normalized with respect to the total production output (i.e., production requirements of the 16 sectors + value added). The resulting normalized column comprising of 17 elements will necessarily add up to 1. Hence, the elements of a column of the I–O matrix can be simultaneously modeled as a Dirichlet distribution, whose formulations are shows below.

The probability density function of the Dirichlet distribution is as follows, where $x_i$ are the Dirichlet random variables, $\alpha_i$ are the associated Dirichlet parameters, and $\Gamma(\cdot)$ is the Gamma function:

$$f(x; \alpha) = \frac{\Gamma\left(\sum_{i=1}^{k} \alpha_i\right)}{\Gamma\left(\prod_{i=1}^{k} \alpha_i\right)} \prod_{i=1}^{k-1} (x_i)^{\alpha_i-1} \left(1 - \sum_{i=1}^{k-1} x_i\right)^{\alpha_k-1}$$
$$0 \leq x_i \leq 1 \text{ and } \alpha_i > 0 \text{ for } i = 1, 2, \ldots, k \tag{6}$$

The general formulation in the previous equation can be rewritten by letting $\alpha_0 = \sum_{i=1}^{k} \alpha_i$.

$$f(x; \alpha) = \frac{\Gamma(\alpha_0)}{\Gamma\left(\prod_{i=1}^{k} \alpha_i\right)} \prod_{i=1}^{k-1} (x_i)^{\alpha_i-1} \left(1 - \sum_{i=1}^{k-1} x_i\right)^{\alpha_k-1} \tag{7}$$

Each proportion $x_i$ in the above Dirichlet distribution can be modeled separately as a marginal distribution of a Gamma distribution variable $y_i$, which follows the following formula.

$$f(y_i; \alpha_i, \alpha_0 - \alpha_i) = \frac{\Gamma(\alpha_0)}{\Gamma(\alpha_i)\Gamma(\alpha_0 - \alpha_i)} y_i^{\alpha_0 - 1} (1 - y_i)^{\alpha_0 - \alpha_i - 1} \tag{8}$$

To derive the values of the Dirichlet variable $x_i$ from the Gamma variable $y_i$, the following algorithm is used.

**Step 1**: For $i = 1, 2, \ldots k$, draw a random number, $y_i$, from a Gamma distribution with parameters $\Gamma(\alpha_i, 1)$, where:

$$\Gamma(\alpha_i, 1) = y_i^{\alpha_i - 1} \frac{e^{-y_i}}{\Gamma(\alpha_i)} \tag{9}$$

**Step 2**: Normalize the realizations from Step 1 to produce the Dirichlet realizations.

$$x_i = \frac{y_i}{\sum_{i=1}^{k} y_i} \tag{10}$$

Figure 1 shows the multiple columns of an I–O matrix, where each column adds up to 1. Each column can be modeled as a Dirichlet distribution using the previous formulations and the foregoing simulation algorithm.

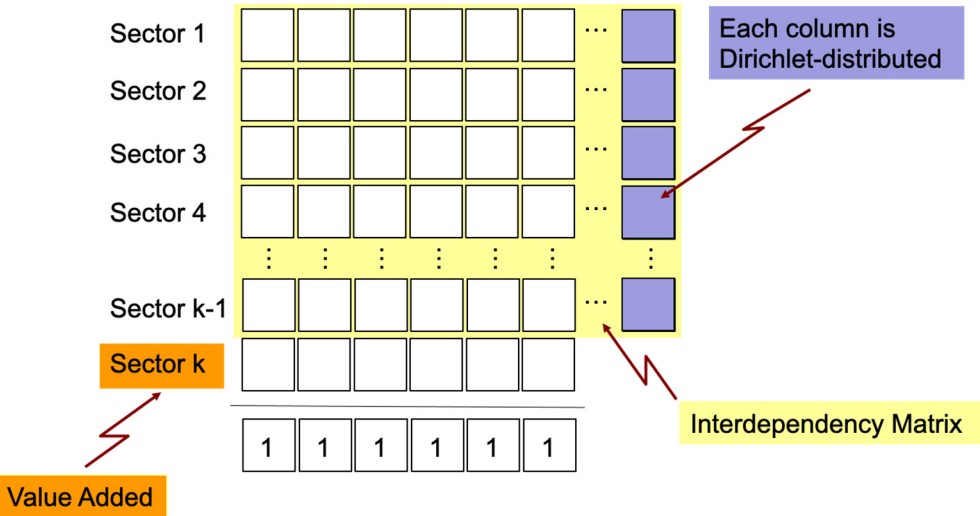

**Figure 1.** Depiction of an I–O matrix where each column is modeled as a Dirichlet distribution.

*2.4. Data Sources*

2.4.1. Input–Output Data (2000, 2006, 2012, 2018)

This study will use the case of the Philippines to illustrate the use of 2000 (National Statistical Coordination Board 2006a), 2006 (National Statistical Coordination Board 2006b), 2012 (Philippine Statistics Authority 2017) and 2018 (Philippine Statistics Authority 2018) Philippine Input–Output Tables. The I–O tables for the different years were re-specified into 16-sector tables that account for one agriculture sector, four industrial sectors and eleven services sectors. Table 1 provides details on the sector disaggregation of the input–output tables that is adapted in the study.

**Table 1.** Sector classification.

| Sector Code | Sector |
|---|---|
| S1 | Agriculture, forestry, and fishing |
| S2 | Mining and quarrying |
| S3 | Manufacturing |
| S4 | Electricity, steam, water and waste management |
| S5 | Construction |
| S6 | Wholesale and retail trade; repair of motor vehicles and motorcycles |
| S7 | Transportation and storage |
| S8 | Accommodation and food service activities |
| S9 | Information and communication |
| S10 | Financial and insurance activities |
| S11 | Real estate and ownership of dwellings |
| S12 | Professional and business services |
| S13 | Public Administration and Defense; Compulsory social security |
| S14 | Education |
| S15 | Human health and social work activities |
| S16 | Other services |

### 2.4.2. Workforce Disruption Scenarios

The COVID-19 pandemic has resulted in significant workforce disruptions across practically every sector of the economy. In this paper, we performed a literature search to determine the workforce disruption percentage in the Philippines. While the disruption fluctuated during the pandemic as influenced by the varying levels of lockdown scenarios and the associated business closures (see Yu et al. 2020a), we will assume an average annualized workforce disruption for simplicity. There were various estimates of workforce disruptions in the Philippines, but the data that have been specifically used in this paper were based on the detailed report of the International Labor Organization (ILO). In its report, it states that: "One quarter of total employment in the Philippines is likely to be disrupted by the impact of COVID-19 on the economy and labour market, either through decreased earnings and working hours or complete job loss" (ILO 2020).

Using the ILO data, given a 25% general workforce disruption, the next step is to determine the workforce dependence of each sector. A labor-intensive sector that is heavily reliant on its workforce is likely to be more severely affected than a sector that is relatively more automated. In the Philippine I–O tables, the sector-specific workforce dependence can be established from the "value added" portion of the data, notably the "compensation of employees". If the total workforce compensation expenditure of a sector is denoted by $w_i$, and its total production output is denoted by $x_i$, then the ratio $w_i/x_i$ gives an estimate of the labor dependence of that sector. Table 2 gives the sector-specific workforce dependence ratios based on 2018 Philippine I–O data.

Hence, the 25% workforce disruption percentage obtained from the ILO is further adjusted to account for the above sector-specific workforce dependence. Furthermore, the baseline scenario assumed in this paper is that the sectors are not able to telework. This baseline is simulated in this paper to determine the savings realized in contrast to the scenario where each sector's workforce disruption rate is further reduced depending on the ability of each sector to telework. The description of such an approach as well as the actual telework multipliers used in the study is described in the next section.

**Table 2.** Ratio of Compensation of Employees with Production Output (both in million PhP[3]).

| Sector Code | Sector | $w_i$ | $x_i$ | $w_i/x_i$ |
|---|---|---|---|---|
| S1 | Agriculture, forestry, and fishing | 580,692 | 3,460,982 | 17% |
| S2 | Mining and quarrying | 23,593 | 245,169 | 10% |
| S3 | Manufacturing | 959,037 | 10,938,541 | 9% |
| S4 | Electricity, steam, water and waste management | 92,438 | 897,255 | 10% |
| S5 | Construction | 576,094 | 3,008,314 | 19% |
| S6 | Wholesale and retail trade; repair of motor vehicles and motorcycles | 975,807 | 4,667,591 | 21% |
| S7 | Transportation and storage | 241,220 | 1,737,302 | 14% |
| S8 | Accommodation and food service activities | 127,419 | 1,074,371 | 12% |
| S9 | Information and communication | 156,760 | 1,051,165 | 15% |
| S10 | Financial and insurance activities | 427,069 | 2,743,564 | 16% |
| S11 | Real estate and ownership of dwellings | 56,837 | 1,530,595 | 4% |
| S12 | Professional and business services | 725,200 | 2,008,912 | 36% |
| S13 | Public Administration and Defense; Compulsory social security | 612,367 | 1,265,078 | 48% |
| S14 | Education | 559,598 | 857,562 | 65% |
| S15 | Human health and social work activities | 84,287 | 460,131 | 18% |
| S16 | Other services | 98,666 | 592,942 | 17% |

### 2.4.3. Telework Data

This study adapts the weighted average teleworkability of occupations by major industry groups in the Philippines based on Generalao (2021). The major industry group were then aggregated based on the sector classification specified in Table 1. Since this study considered the proportion of the occupations that are not teleworkable (a dimensionless measure), Table 3 presents the information on non-teleworkable jobs, $(1 - T_i)$, where $T_i$ is the proportion of teleworkable jobs for sector $i$ as adapted from Generalao (2021).

**Table 3.** Proportion of non-teleworkable occupations by major industry group.

| Sector Code | Sector | $(1 - T_i)$ |
|---|---|---|
| S1 | Agriculture, forestry, and fishing | 0.9191 |
| S2 | Mining and quarrying | 0.8744 |
| S3 | Manufacturing | 0.8257 |
| S4 | Electricity, steam, water and waste management | 0.7252 |
| S5 | Construction | 0.9616 |
| S6 | Wholesale and retail trade; repair of motor vehicles and motorcycles | 0.6058 |
| S7 | Transportation and storage | 0.8038 |
| S8 | Accommodation and food service activities | 0.8332 |
| S9 | Information and communication | 0.3571 |
| S10 | Financial and insurance activities | 0.2264 |
| S11 | Real estate and ownership of dwellings | 0.4763 |
| S12 | Professional and business services | 0.4987 |
| S13 | Public Administration and Defense; Compulsory social security | 0.6366 |
| S14 | Education | 0.4327 |
| S15 | Human health and social work activities | 0.7009 |
| S16 | Other services | 0.7096 |

## 3. Results

### 3.1. Processing the I–O Data for Uncertainty Analysis

To be able to estimate the Dirichlet parameters, the values in the I–O matrix were normalized to compute for the shares for each sector. To get the shares for each sector, we first calculated the total input of each sector, $S_{it}$ for each sector $i$ for each year $t$ and it includes the value added, $VA_{it}$ (see Equation (11)). We let $I_{it}$ be the intermediate input of sector $i$ and at year $t$. The value added, $VA_{it}$, includes: (1) Compensation of Employees;

(2) Consumption of Fixed Capital; (3) Indirect Taxes less subsidies; and (4) Operating Surplus[4]. There are 16 sectors in the Philippine I–O table and the study uses the I–O tables for the years 2000, 2006, 2012, and 2018.

$$S_{it} = \sum_{i=1}^{16} I_{it} + \sum VA_{it} \tag{11}$$

To get the share of each intermediate input for each sector $i$ at year $t$, we take the value of the sector's intermediate input and divide it by the total input of each sector, $S_{it}$ (see Equation (12)). The share of the value added is also calculated similarly as shown in Equation (13). The estimated shares are then used as the dataset for the uncertainty analysis and the Dirichlet parameters.

$$Share\ of\ intermediate\ input\ of\ sector\ i\ at\ year\ t = \frac{I_{it}}{S_{it}} \tag{12}$$

$$Share\ of\ value\ added\ of\ sector\ i\ at\ year\ t = \frac{\sum VA_{it}}{S_{it}} \tag{13}$$

### 3.2. Dirichlet Parameters

The statistical distribution of the values in the I–O matrix can be generated with beta or gamma distribution using the Dirichlet parameter $\hat{a}_{ij}$ and $\hat{\beta}_{ij}$ using the following equations:

$$\alpha_{ij} = \mu_{ij}\left(\frac{\mu_{ij} - \mu_{ij}^2}{\sigma_{ij}^2} - 1\right) \tag{14}$$

$$\beta_{ij} = (1 - \mu_{ij})\left(\frac{\mu_{ij} - \mu_{ij}^2}{\sigma_{ij}^2} - 1\right) \tag{15}$$

$$\hat{a}_{ij} = \mu_{ij}\left(\alpha_{kj} + \beta_{kj}\right) \tag{16}$$

$$\hat{\beta}_{ij} = \sum_{i=1}^{k} \hat{\alpha}_{ij} - \hat{\alpha}_{ij} \tag{17}$$

where $\mu_{ij}$ and $\sigma_{ij}$ are the mean and standard deviation of the entry in the I–O matrix and $\alpha_{kj}$ is generated using the following equation:

$$\alpha_{kj} = 1 - \sum_{i=1}^{k-1} \alpha_{ij} \tag{18}$$

Different realizations with the I–O matrix can be generated with beta distribution using the parameters $\hat{\alpha}_{ij}$ and $\hat{\beta}_{ij}$ or with Gamma distribution using $\hat{\alpha}_{ij}$ with a scale of 1 and then normalization to be consistent with the property of the I–O matrix (see Supplementary Materials). Both distributions provide the same expected value and probability distribution. The generation of these parameters is implemented using Numpy' gamma and beta sampling functions (random.gamma and random.beta) (Harris et al. 2020).

### 3.3. Description of Scenarios and Visualization of Results

This study considers two scenarios: (1) no telework, and (2) with telework. Scenario 1 does not allow for telework in any sector. This means that the full impact of the initial disruption ($c$*) will ripple through the different sectors in the economy. Scenario 2 accounts for teleworkability of jobs in the Philippines based on Generalao (2021). With telework, the initial disruption ($c$*) is reduced by the proportion of jobs that are teleworkable, such that the initial disruption in Scenario 2 for each sector $i$ is defined as $c_i^* * (1 - T_i)$, where $T_i$ is the teleworkability index of sector $i$. On the other hand, $c_i^*$ is the sector-specific workforce disruption, which is computed as the ILO's 25% general workforce disruption multiplied with the sector-specific $w_i/x_i$ ratio found on Table 2.

The losses from different simulation runs are computed based on different realizations of the I–O matrix; a boxplot can then be generated showing the uncertainty of these losses. Figure 2 shows the boxplots for the percent losses and output losses for 25% disruption to the general workforce. The plot shows that the largest loss is incurred in the manufacturing sector with around PhP 1.9 trillion to PhP 3.4 trillion followed by the wholesale and trade sector with PhP 500 billion to PhP 700 billion in losses. The plots also show that these two sectors show the highest variability in terms of economic losses. Both sectors maintain their ranks except if the highest loss of agricultural sector is compared with the lowest loss of the wholesale and trade sector. It shows that if the agricultural sector is affected by the workforce disruption worse than expected, it can incur losses comparable to other sectors. It is also evident in Figure 2 when the percentage loss in agricultural sector is higher than in wholesale and trade. The education sector has the second highest percentage loss to manufacturing and is one with the least variability among sectors.

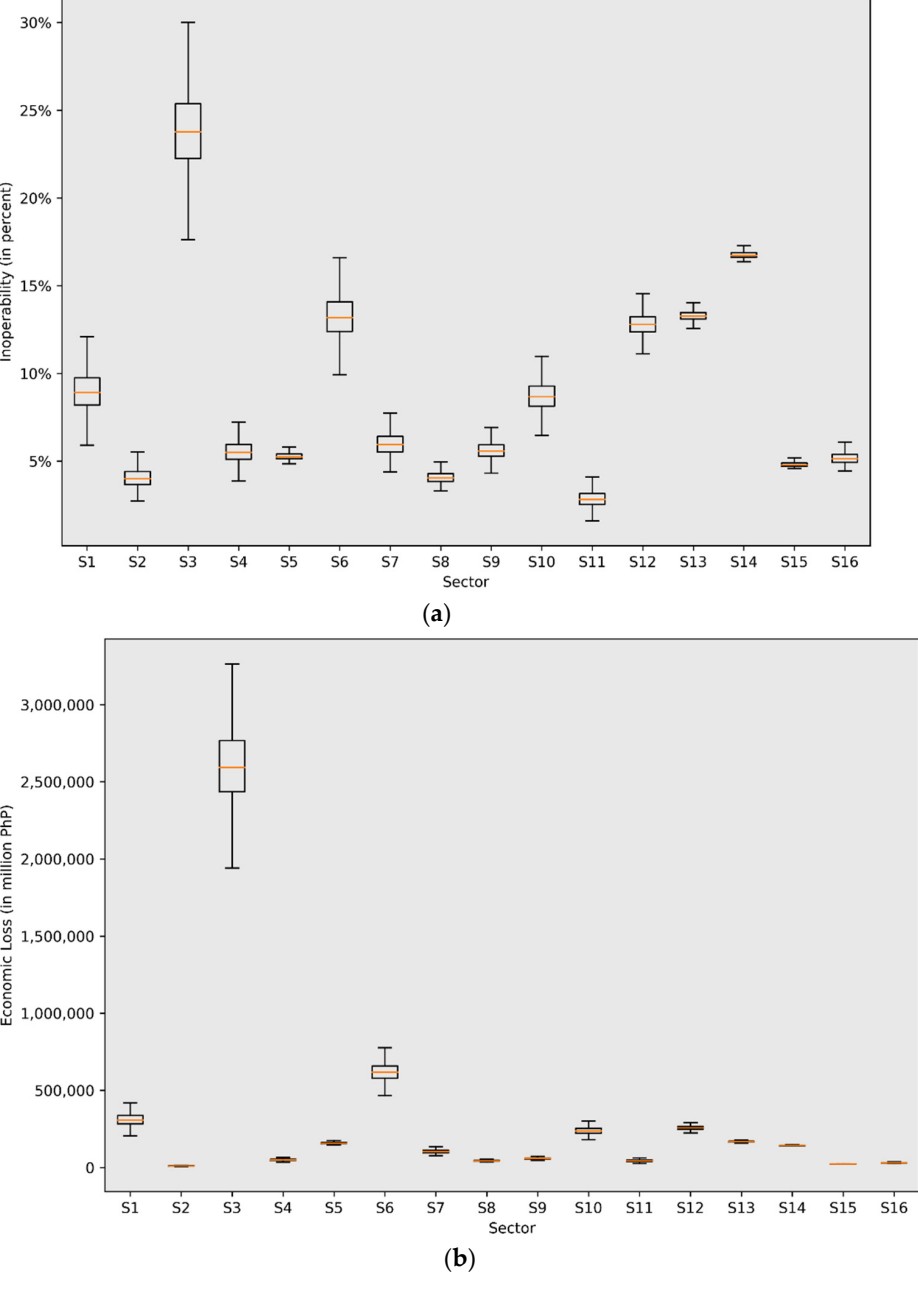

**Figure 2.** Losses incurred to different sector due to 25% disruption in general workforce in terms of (**a**) percent losses and (**b**) actual economic loss assuming absence of telework performance.

Figure 3 shows the boxplots for the percent losses and output losses for 25% disruption considering the telework data. The actual losses are generally lower than that when the telework data are not considered, providing concrete evidence that teleworking as a resilience strategy is effective in curbing the economic losses relative to the baseline case (i.e., no telework scenario). The same trend is observed except for certain sectors such as the transportation sector, and education sector wherein the losses in the education sector are lower than the transportation sector if the telework performance is considered. The rank reverses when the telework data are not considered in the computation.

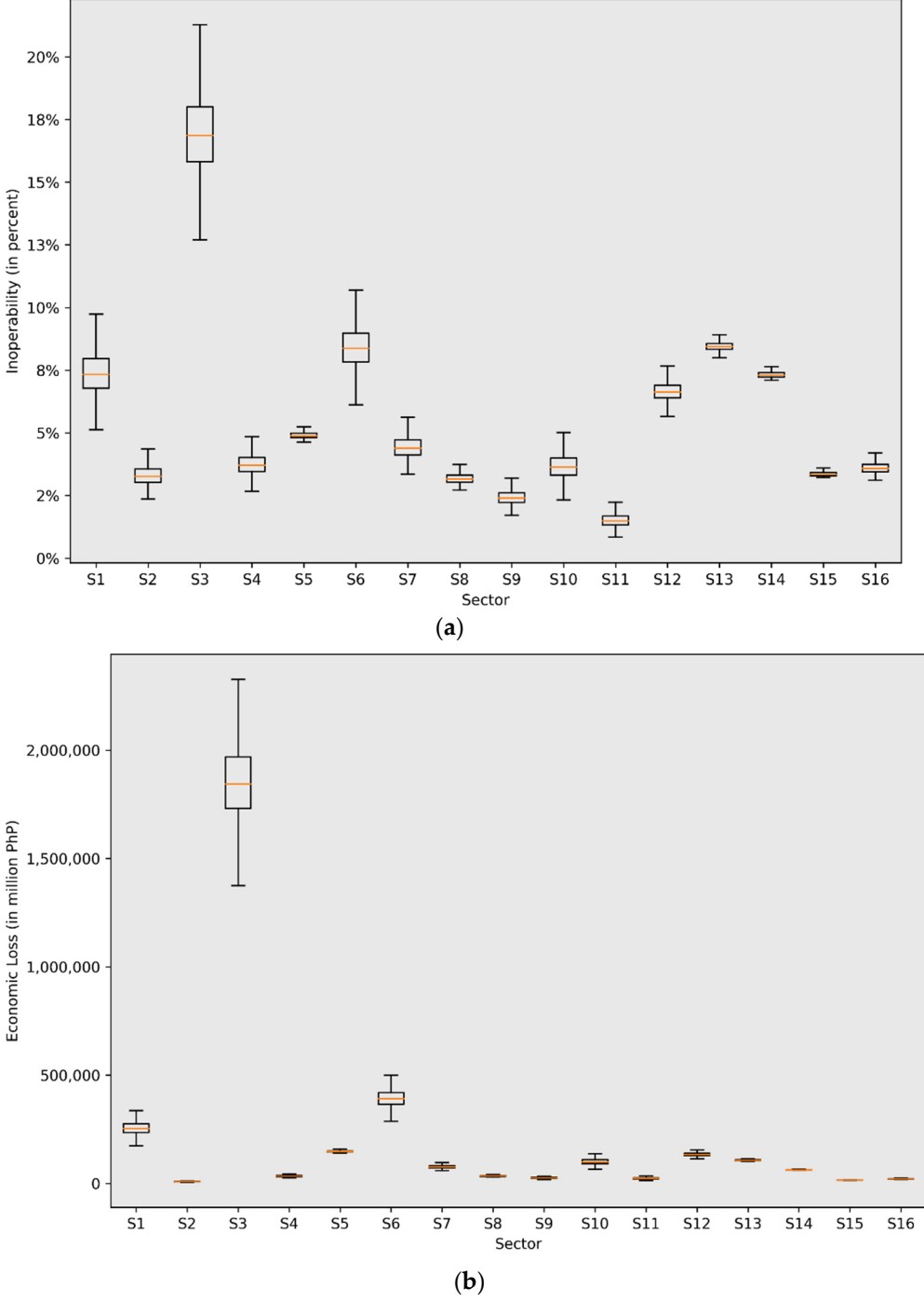

**Figure 3.** Losses incurred to different sectors due to 25% disruption in the general workforce in terms of (**a**) percent losses, and (**b**) actual economic loss considering the sector-specific telework data.

## 4. Discussion

### 4.1. Specific Findings on Philippine I–O Sectors

The results of the uncertainty analysis show how allowing telework in the 16 sectors presented has significantly decreased the estimated economic losses due to the COVID-19 pandemic in the Philippines. If the telework arrangements were not adopted, the estimates show that the country will have a substantial economic loss which is equivalent to PhP 4.89 trillion, or approximately USD 86.9 billion due to the lockdowns. All 16 sectors benefitted from allowing the telework arrangement as the losses with telework was significantly decreased from PhP 4.89 trillion to PhP 3.3 trillion, or approximately USD 58.7 billion.

Three sectors benefited the most from reducing the economic losses by allowing telework during the COVID-19 pandemic. The first sector, which is the manufacturing sector (S3), had the biggest difference between with and without telework, with the difference amounting to PhP 750 billion (USD 13.5 billion). It is expected, however, that out of all the sectors, the manufacturing sector will have the most losses since most of its economic activities require the physical presence of laborers. The second sector that also benefitted from allowing telework was the wholesale and retail trade (S6)[5]. The difference in economic loss between with and without telework is estimated at PhP 226 billion (USD 4 billion). The third sector that had significantly lower losses with the introduction of telework was the financial and insurance activities (S10). The estimated difference in the economic losses with and without telework is PhP 139 billion (USD 2.463 billion). Looking at the rest of the sectors, the trend is the same in which having telework has decreased the estimated economic losses due to the pandemic.

An interesting insight from the results is that there are non-teleworkable sectors, as classified by Gaduena et al. (2022) and Generalao (2021), that still benefitted indirectly from teleworking. These industries include agriculture, forestry, and fishing (S1), manufacturing (S3), wholesale and retail trade and repair of motor vehicles and motorcycles (S6), transportation and storage (S7), and public administration and defense and compulsory social security (S13). Although these sectors mostly require a physical presence, the estimates suggest that introducing telework can mitigate the economic losses due to the pandemic. This effect can be attributed to the use of teleworking arrangements both within these sectors, and in other sectors with which they have strong supply chain linkages.

### 4.2. Uncertainty Analysis

Tables 4–6 shows the comparison between economic losses in million PhP considering the 25th, 50th and 75th percentiles in the Dirichlet distribution for each sector. Policy makers can use these percentile estimates in accordance with their risk appetite. In all percentile points, most sectors such as the manufacturing, wholesale and trade, and agriculture sectors maintain their ranks whether the telework data are considered or not. Changes in the ranking can be observed at certain sectors when the percentile is changed. For instance, the larger variability in financial and insurance activities sector allows it to outrank the public administration and social security sectors when the losses are changed from the 50th to the 75th percentile. This change shows that variability should be considered in prioritizing certain sectors when workforce disruptions are experienced.

The results generated by the simulations are consistent and in the same order of magnitude as the estimates published by the National Economic and Development Authority. In particular, the agency has estimated the total economic losses associated with COVID-19 business interruptions and quarantines for the year 2020 to be at PhP 4.3 trillion (National Economic and Development Authority 2021). A case in point, our 25th percentile estimates (see Table 4) range from PhP 3.1 trillion (with telework) to PhP 4.6 trillion (without telework). Furthermore, our median estimates (see Table 5) range from PhP 3.3 trillion (with telework) to PhP 4.8 trillion (without telework). Finally, the 7th percentile estimates of economic losses in Table 6 range from PhP 3.5 trillion (with telework) to PhP 5.2 trillion (without telework). The total economic losses shown at the bottom of Tables 4–6 consistently capture the PhP 4.3 trillion estimate published by the National Economic and

Development Authority (2021), hence providing a validation of the model results generated in our simulations.

**Table 4.** Economic losses for different sectors based on the 25th percentile in the Dirichlet distribution.

| Sector Code | Sector | No Telework | With Telework | Difference |
|---|---|---|---|---|
| S1 | Agriculture, forestry, and fishing | 282,719 | 234,678 | 48,041 |
| S2 | Mining and quarrying | 8963 | 7389 | 1574 |
| S3 | Manufacturing | 2,437,518 | 1,728,333 | 709,185 |
| S4 | Electricity, steam, water, and waste management | 45,959 | 30,971 | 14,988 |
| S5 | Construction | 154,412 | 144,782 | 9630 |
| S6 | Wholesale and retail trade; repair of motor vehicles and motorcycles | 577,483 | 365,226 | 212,257 |
| S7 | Transportation and storage | 96,076 | 71,497 | 24,579 |
| S8 | Accommodation and food service activities | 41,300 | 32,588 | 8712 |
| S9 | Information and communication | 55,598 | 23,287 | 32,311 |
| S10 | Financial and insurance activities | 223,999 | 90,589 | 133,410 |
| S11 | Real estate and ownership of dwellings | 38,782 | 20,281 | 18,501 |
| S12 | Professional and business services | 248,541 | 128,356 | 120,185 |
| S13 | Public Administration and Defense; Compulsory social security | 165,688 | 105,437 | 60,251 |
| S14 | Education | 142,433 | 62,043 | 80,390 |
| S15 | Human health and social work activities | 21,650 | 15,134 | 6516 |
| S16 | Other services | 29,239 | 20,423 | 8816 |
| | Total | 4,570,360 | 3,081,014 | 1,489,346 |

**Table 5.** Economic losses for different sectors based on the 50th percentile (median) in the Dirichlet distribution.

| Sector Code | Sector | No Telework | With Telework | Difference |
|---|---|---|---|---|
| S1 | Agriculture, forestry, and fishing | 307,765 | 253,668 | 54,097 |
| S2 | Mining and quarrying | 9804 | 7981 | 1823 |
| S3 | Manufacturing | 2,599,304 | 1,847,460 | 751,844 |
| S4 | Electricity, steam, water, and waste management | 49,449 | 33,269 | 16,180 |
| S5 | Construction | 157,967 | 146,988 | 10,979 |
| S6 | Wholesale and retail trade; repair of motor vehicles and motorcycles | 614,992 | 390,473 | 224,519 |
| S7 | Transportation and storage | 102,952 | 76,081 | 26,871 |
| S8 | Accommodation and food service activities | 43,406 | 33,846 | 9560 |
| S9 | Information and communication | 58,663 | 25,070 | 33,593 |
| S10 | Financial and insurance activities | 238,084 | 99,343 | 138,741 |
| S11 | Real estate and ownership of dwellings | 43,137 | 22,751 | 20,386 |
| S12 | Professional and business services | 256,613 | 133,108 | 123,505 |
| S13 | Public Administration and Defense; Compulsory social security | 167,897 | 106,825 | 61,072 |
| S14 | Education | 143,404 | 62,630 | 80,774 |
| S15 | Human health and social work activities | 21,996 | 15,361 | 6635 |
| S16 | Other services | 30,441 | 21,186 | 9255 |
| | Total | 4,845,874 | 3,276,040 | 1,569,834 |

**Table 6.** Economic losses for different sectors based on the 75th percentile in the Dirichlet distribution.

| Sector Code | Sector | No Telework | With Telework | Difference |
|---|---|---|---|---|
| S1 | Agriculture, forestry, and fishing | 337,089 | 276,171 | 60,918 |
| S2 | Mining and quarrying | 10,799 | 8674 | 2125 |
| S3 | Manufacturing | 2,780,086 | 1,972,136 | 807,950 |
| S4 | Electricity, steam, water, and waste management | 53,585 | 35,977 | 17,608 |
| S5 | Construction | 162,628 | 149,988 | 12,640 |
| S6 | Wholesale and retail trade; repair of motor vehicles and motorcycles | 656,070 | 419,259 | 236,811 |
| S7 | Transportation and storage | 111,320 | 81,573 | 29,747 |
| S8 | Accommodation and food service activities | 46,200 | 35,549 | 10,651 |
| S9 | Information and communication | 62,338 | 27,358 | 34,980 |
| S10 | Financial and insurance activities | 254,440 | 109,449 | 144,991 |
| S11 | Real estate and ownership of dwellings | 48,309 | 25,830 | 22,479 |
| S12 | Professional and business services | 265,813 | 138,767 | 127,046 |
| S13 | Public Administration and Defense; Compulsory social security | 170,319 | 108,397 | 61,922 |
| S14 | Education | 144,715 | 63,465 | 81,250 |
| S15 | Human health and social work activities | 22,499 | 15,700 | 6799 |
| S16 | Other services | 31,966 | 22,234 | 9732 |
| | Total | 5,158,176 | 3,490,527 | 1,667,649 |

## 5. Conclusions

The COVID-19 pandemic has brought an unprecedented cascade of economic losses across the globe. The largest contributor to economic loss is attributable to workforce disruptions resulting from quarantines and business closures. Workforce disruptions subsequently led to significant direct and indirect ripple effects on the operation of the economic sectors. Several studies, even prior to the COVID-19 pandemic, already indicated that business interruptions during disasters were a top factor leading to economic losses. In this paper, we perform an ex-post analysis of the COVID-19 workforce disruptions in the Philippines. The aim is to estimate the workforce-induced economic losses for year 2020. Using a series of Philippine I–O data, we estimated the probability distributions of the model parameters to allow for uncertainty analysis in the estimates of economic losses. The simulation results are consistent with the official deterministic figure of 4.3 trillion PhP. Furthermore, a novel feature of this paper is the use of Dirichlet distributions to generate a range of economic losses (e.g., 25th, 50th, and 75th percentiles) for two scenarios: (i) an extreme case assuming no telework, and (ii) sector-specific telework analysis using estimates from previous studies. The advantage of uncertainty analysis is that it allows the presentation of results as a distribution, consistent with the argument that a fixed-point estimate often creates an illusion of precision, which can restrict the flexibility in resource allocation decisions. A range of estimates allows for the implementation of "what if" scenarios, which leads to a more holistic and robust policymaking.

Some of the highlights of the results in this paper are as follows: Intuitively, the magnitude of losses depends on the size of the sector (GDP), as well as the labor-dependence of the sectors. To wit, sectors that contribute highly to GDP were among the most affected sectors such as manufacturing, trade, agriculture, and construction, which happen to be among the highest contributors to the Philippine GDP. It has also been found that labor-dependent sectors with less telework index can be observed in the fractional loss (inoperability) rankings albeit their moderate contribution to the GDP, such as public administration, education, professional and business Services.

To conclude, current disaster management policies need to be enhanced to minimize the impact of pandemics and future disasters. The COVID-19 pandemic in particular has exposed challenges and constraints in socioeconomic and infrastructure systems. Nonpharmaceutical intervention measures could reduce the impact on the workforce, healthcare systems, and continuity of government. Measures like teleworking can have a significant

impact on the extent to which the pandemic curve can be flattened. The benefits of such arrangements were shown to also indirectly accrue to sectors that are not inherently suited to teleworking. Results from this paper can be used to determine the key sectors that are most impacted by the disaster and can be used to formulate polices to target sectors that contribute the most to the direct and ripple effects of indirect losses. These results can be generalized to other countries in preparation for future pandemics. Finally, the methodology for this paper can be adopted and extended for other disaster risk management studies, particularly those that affect the workforce more than the infrastructure.

**Supplementary Materials:** The 16-sector Philippine Input–Output Tables used in this study can be accessed through 10.6084/m9.figshare.20374587. The Python code used for this study can be accessed in https://github.com/fredtapia/IO-Dirichlet (accessed on 17 August 2022).

**Author Contributions:** Conceptualization, J.R.S.; methodology, J.R.S., A.L. and J.F.D.T.; software, J.F.D.T.; validation, J.R.S. and K.D.S.Y.; formal analysis, J.R.S., A.L. and J.F.D.T.; investigation, J.R.S., J.F.D.T. and K.B.A.; resources, K.D.S.Y.; data curation, K.D.S.Y.; writing—original draft preparation, J.R.S. and C.A.S. writing—review and editing, C.A.S., R.R.T. and K.B.A.; visualization, J.F.D.T.; investigation, K.D.S.Y.; supervision, J.R.S. and K.D.S.Y.; project administration, J.R.S.; funding acquisition, J.R.S. and R.R.T. All authors have read and agreed to the published version of the manuscript.

**Funding:** This research was funded by the National Science Foundation (NSF) grant number 1832635.

**Acknowledgments:** We acknowledge the support from the National Science Foundation (Award #1832635), Fulbright US Scholar Program, and Philippine-American Educational Foundation (PAEF).

**Conflicts of Interest:** The authors declare no conflict of interest.

## Notes

1.   COVID-19, or coronavirus disease 2019, a contagious disease caused by a virus, the severe acute respiratory syndrome coronavirus 2 (SARS-CoV-2).
2.   In the field of epidemiology, the basic reproduction number ($R_0$) is a parameter used to describe the expected number of people that can be infected by a sick individual. It is used extensively in the mathematical formulations for the "Susceptible Infected Removed" (SIR) model and extensions (Anderson and May 1992).
3.   For reference, USD 1 is equivalent to approximately PhP 50 in year 2020.
4.   It should be noted that for the 2018 IO, the Operating Surplus is not included. The Operating Surplus is removed from the IO matrices.
5.   In the grouping of the industries in the Philippine IO, wholesale and retail trade is combined with repair of motor vehicles and motorcycles.

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
