# Peer review of "Uncertainty Analysis of Business Interruption Losses in the Philippines Due to the COVID-19 Pandemic"

_economies, doi:10.3390/economies10080202_

Round 1

Reviewer 1 Report

Manuscript

Title:Uncertainty Analysis of Pandemic-Induced Business Interruption Losses in the Philippines”

Authors: ??????

Dear Authors

I revised the manuscript: "Uncertainty Analysis of Pandemic-Induced Business Interruption Losses in the Philippines " submitted to the “Economies” Journal. The paper is very interesting. However, I have some concerns, which need to be addressed.

Line 2-3. Article topic

The topic of the article is formulated with sufficient precision and contains the essential components of research

Abstract

Line 4-16.

Line 4, 6. „COVID-19”

Please explain the introduced abbreviations of terms, possibly as soon as they are introduced in the content of the article. A single explanation is sufficient in the execution of the explanation.

For example: „COVID-19 - coronavirus disease 2019, a contagious disease caused by a virus, the severe acute respiratory syndrome coronavirus 2 (SARS-CoV-2)

The abstract does not include a description of the research results and conclusions that immanently result from the research. It is possible to give a brief overview of the research results but an excessive generalisation may indicate a lack of research results that are interesting to the reader. Please consider indicating the most important result and the most important conclusion reached as a result of the research. 

Keywords. Line 17-18. Keywords are a mix of detailed and general phrases. In my opinion, the set of keywords can be optimised. Please consider this.

1. INTRODUCTION

The content of the chapter correctly introduces the reader to the problems of the article. The quite abbreviated form of the statement contributes to limiting the topics of the described problem. Please take this into account. The presentation of the goal and scope of the work is residual and should be better described.    

Line 24."....the1918 H1N1 pandemic...." The abbreviation should not be deprived of explanation in a situation of thematically convergent specialist information. 

Line 26, 37, 40, 42, 50, 54, 59, 61, 68, 77, 79. „…(Taubenberger and Morens 2006)….” Authors do not use notation of literature items by assigned ordinal numbers. Please verify the method of notation with publication requirements.

Line 27. „….1952 H2N2, 1968 H3N2….” The abbreviation should not be deprived of explanation in a situation of thematically convergent specialist information. 

Line 27. "...H1N1pdm2009...."  The notation without spaces is incomprehensible. Please standardise the notation of this type of information in the content of the article.

Line 78. „…EU…” Please explain the introduced abbreviations of terms, possibly as soon as they are introduced in the content of the article. A single explanation is sufficient in the execution of the explanation.

2. MATERIALS AND METHODS

The division of the chapter content into subsections is logical and reflects well the characteristics of the presented information.

Designations Eq. 4, Eq. 5, Eq. 6, Eq. 7, Eq. 8, Eq. 9, Eq. 10. Please complete the references to the presented formulas (equations) in the text of the chapter. 

Line 93, 95, 96, 103, 117, 118, 122, 128, 131, 134, 137, 139, 142, 151, 154, 155, 159, 162, 166, 169, 170, 172, 176, 178, 185, 204, 207, 210, 215, 236, 245, 247, 255, 256, 303, 304, 305, 318, 324, 347, 351, . „…(Okuyama and Santos 2014))….”

Authors do not use notation of literature items by assigned ordinal numbers. Please verify the method of notation with publication requirements.

Line 144: '...NPI...'. Is "NPI" the singular of the phrase and should the abbreviation be understood as "non-pharmaceutical intervention"? Please complete the description of the abbreviation in the text of the article.

Line 165 "OECD". Although the quoted abbreviation is part of the coding of the publication as the author of the publication, the abbreviation should be explained due to the form of citation in the text.

Please explain the introduced abbreviations of terms, possibly as soon as they are introduced in the content of the article. A single explanation is sufficient in the execution of the explanation.

Line 188. Mathematical Formula (Equation) No. 1. The notation „x” is simultaneously present as the dependent variable and the independent variable? Please verify the correctness of the notation for the relationship of factors presented. 

Line 212 Mathematical Formula (Equation) No. 3. The notation „q” is simultaneously present as the dependent variable and the independent variable? Please verify the correctness of the notation for the relationship of factors presented. 

Line 248, 255, 301, 306. „….I-O…..” Please explain the introduced abbreviations of terms, possibly as soon as they are introduced in the content of the article. A single explanation is sufficient in the execution of the explanation.

Line 258 Formula (Equation) 5. The description of the components of the mathematical formula (equation) is insufficient. There is a lack of explanations of the variables (for example: ?, β, Γ, xi, yi). Please complete this.

Line 260, 523, 524, 526, 528 „….GDP….” Please explain the introduced abbreviations of terms, possibly as soon as they are introduced in the content of the article. A single explanation is sufficient in the execution of the explanation.

Line 271. The description of the components of the mathematical formula (equation) is insufficient. There is a lack of explanations of the variables (for example: ?, β, Γ, xi, yi). Please complete this.

Line 275. The description of the components of the mathematical formula (equation) is insufficient. There is a lack of explanations of the variables (for example: ?, β, Γ, xi, yi). Please complete this.

Line 280. The description of the components of the mathematical formula (equation) is insufficient. There is a lack of explanations of the variables (for example: ?, β, Γ, xi, yi). Please complete this.

Line 287. The description of the components of the mathematical formula (equation) is insufficient. There is a lack of explanations of the variables (for example: ?, β, Γ, xi, yi). Please complete this.

Line 290. The description of the components of the mathematical formula (equation) is insufficient. There is a lack of explanations of the variables. Please complete this.

Line 301. "...I-O Data (2000, 2006, 2012, 2018)...." If the authors indicate a record of years and intervals of time then the description should include the term "years of research". This will make the information unambiguous.

Line 331, 335 Table 2. „….ratio ??/?? gives…..” The notation with a slash is colloquial and incomprehensible. Please explain clearly the intention of this coding.

Please use the notation of the quotient in units of measure using mathematical notation with a power exponent for example: wi·xi-1

Line 335. „PhP….” Please explain the introduced abbreviations of terms, possibly as soon as they are introduced in the content of the article. A single explanation is sufficient in the execution of the explanation.

Line 335 Table 2. „wi”, „xi” . Lack of units of measurement assigned to numerical values.Given values require an unambiguous reference to the unit of measurement. Dimensionless values also need to be indicated, for example "[-]" The unit of measure as the country's currency (PhP?) needs to be clearly described and explained in the text of the article.

Line 353 Table 3.  Lack of units of measurement assigned to numerical values.Given values require an unambiguous reference to the unit of measurement. Dimensionless values also need to be indicated, for example "[-]"

3. RESULTS

The division of the chapter content into subsections is logical and reflects well the characteristics of the presented information.

Line 389, 394 . „…(Harris et al., 2020)….” Authors do not use notation of literature items by assigned ordinal numbers. Please verify the method of notation with publication requirements.

Designations Eq. 11, Eq. 12, Eq. 13, Eq. 14, Eq. 15, Eq. 16, Eq. 17, Eq. 18. Please complete the references to the presented formulas (equations) in the text of the chapter. 

Line 404. „…1.9 trillion to 3.4 trillion…..” Please indicate the unit of measurement for the numerical values. Please indicate the unit of measure for each of the two range values.

Line 405. "..... 500 to 700 billion....." Please indicate the unit of measurement for the numerical values. Please indicate the unit of measure for each of the two range values.

Line 425 Figure 2 b) "...in million Php...". PhP or Php? Please standardise the method of notation for describing the axes of the graph.

Line 435 Figure 3 (b) "...in million Php...". PhP or Php? Please standardise the method of notation for describing the axes of the graph.

4. DISCUSSION

The division of the chapter content into subsections is logical and reflects well the characteristics of the presented information.

The discussion of the results is conducted based on only 3 items of the relevant literature. Please explain in the text of the chapter the reason for such a limited form of expression in the key point of the work. The lack of additional information significantly limits the scientific level of the presented content.

Line 446 and similar in the chapter. "....USD 86.9 billion...." The notation of the unit of measure is not homogeneous - , $ and USD. Please unify the notation of the unit of measure in this area.

Line 485 Table 4. Given values require an unambiguous reference to the unit of measurement. Dimensionless values also need to be indicated, for example "[-]"

Line 464, 495 . „…Gaduena et al. (2022) and Generalao (2021)….” Authors do not use notation of literature items by assigned ordinal numbers. Please verify the method of notation with publication requirements.

Line 487 Table 5. Given values require an unambiguous reference to the unit of measurement. Dimensionless values also need to be indicated, for example "[-]"

Line 489 Table 6. Given values require an unambiguous reference to the unit of measurement. Dimensionless values also need to be indicated, for example "[-]"

5. CONCLUSIONS

The authors' conclusions summarise all the information and results presented in the article. It should be noted, however, that the lack of reference to assigned tables and figures to the conclusions, the lack of sequential numbers of the commented information, makes it difficult for the readers to follow and identify with the course of the authors' statements. The conclusions need to be modified due to the requirement to better structure the statements of the chapter.

Reviewer 2 Report

Uncertainty Analysis of Pandemic-Induced Business Interruption Losses in the Philippines

I find it a very interesting and timely article. I am going to make some comments about it.

1It is necessary to show more the relevance and novelty of this work, in relation to the existing literature.

2-The title does not seem to me to reflect exactly the objectives of the work, I think it should be modified in this sense

3.The references related to the input-output model and the IIM should be improved, they are few

4 I consider it more appropriate that section 2.1 Overview of Pandemic Impact Modeling, not be within section 2, but rather that it be a separate section and prior to Materials and Methods section

5 Further motivate the use of the Dirichlet distribution in this paper
